# Human Tongue Electrophysiological Response to Oleic Acid and Its Associations with PROP Taster Status and the *CD36* Polymorphism (*rs1761667*)

**DOI:** 10.3390/nu11020315

**Published:** 2019-02-01

**Authors:** Giorgia Sollai, Melania Melis, Mariano Mastinu, Danilo Pani, Piero Cosseddu, Annalisa Bonfiglio, Roberto Crnjar, Beverly J. Tepper, Iole Tomassini Barbarossa

**Affiliations:** 1Department of Biomedical Sciences, University of Cagliari, Monserrato, CA 09042, Italy; gsollai@unica.it (G.S.); melaniamelis@unica.it (M.M.); mariano.mastinu@unica.it (M.M.); crnjar@unica.it (R.C.); 2Department of Electrical and Electronic Engineering, University of Cagliari, Piazza d’Armi, Cagliari 09123, Italy; pani@diee.unica.it (D.P.); piero.cosseddu@diee.unica.it (P.C.); annalisa@diee.unica.it (A.B.); 3Department of Food Science, School of Environmental and Biological Sciences, Rutgers University, New Brunswick, NJ 08901-8520, USA; btepper@sebs.rutgers.edu

**Keywords:** electrophysiological recording from human tongue, fat perception, *CD36*, PROP tasting

## Abstract

The perception of fat varies among individuals and has also been associated with *CD36 rs1761667* polymorphism and genetic ability to perceive oral marker 6-n-propylthiouracil (PROP). Nevertheless, data in the literature are controversial. We present direct measures for the activation of the peripheral taste system in response to oleic acid by electrophysiological recordings from the tongue of 35 volunteers classified for PROP taster status and genotyped for *CD36*. The waveform of biopotentials was analyzed and values of amplitude and rate of potential variation were measured. Oleic acid stimulations evoked positive monophasic potentials, which represent the summated voltage change consequent to the response of the stimulated taste cells. Bio-electrical measurements were fully consistent with the perceived intensity during stimulation, which was verbally reported by the volunteers. ANOVA revealed that the amplitude of signals was directly associated, mostly in the last part of the response, with the *CD36* genotypes and PROP taster status (which was directly associated with the density of papillae). The rate of potential variation was associated only with *CD36,* primarily in the first part of the response. In conclusion, our results provide direct evidence of the relationship between fat perception and *rs1761667* polymorphism of the *CD36* gene and PROP phenotype.

## 1. Introduction

Over the last decade, multiple effects of dietary fatty acids as regulators of lipid and energy metabolism in human health and disease outcomes have been pointed out [1]. Therefore, the capability to distinguish them in a diet can have important nutritional implications for the health of volunteers, and studies aiming to analyze the fat perception and discrimination are important to understand the mechanisms involved in the choice of fat-rich foods [2].

Dietary fats were traditionally thought to have no ‘taste’ of their own, but rather to be sensed through their textural and odorant properties [3]. Emerging findings have since disputed this understanding by demonstrating an important involvement of taste in fat detection, which has been proposed as a sixth primary taste quality [2,3,4,5,6]. Although dietary lipids are mostly triglycerides, free long-chain fatty acids released from dietary lipids during oral processing seem to be accountable for fat taste perception [7,8]. In fact, the cleavage of triglycerides into free fatty acids by a lingual lipase has been shown both in rodents [9] and in humans [5]. Various classes of fatty acid receptors have been proposed for the taste transduction of lipids [10,11], including the multifunctional CD36 scavenger receptor [11,12,13,14,15,16], which is primarily responsible for the detection of long chain fatty acids on the tongue [5,17,18]. CD36 is a membrane glycosylated protein whose expression is controlled by the *CD36* gene and regulated by its allelic diversity [19]. The exchange of A for G in the *rs1761667* single nucleotide polymorphism (SNP) has been shown to decrease protein expression [19], and is associated with a reduced oral ability to perceive fatty acids [5,20,21]. Ethnic-specific effects were also observed in one experiment where East Asians, but not Caucasians, with the AA genotype showed a reduced ability to perceive fatty acids [22]. The substitution of A for G in this SNP has also been shown to influence fat preference [23]. In addition, recent results suggest that this SNP is differentially related with body composition and endocannabinoid levels in lean and obese volunteers [24].

Some studies have also shown that changes in taste sensitivity to fats could be related to differences in general taste sensitivity as indicated by variations in the salivary protein expression of gustin (carbonic anhydrase VI), which has been associated with both taste perception [25], density of papillae and function [26,27]. In addition, it is known that sensitivity to and preference for fat has also been associated with the genetic ability to perceive the oral marker stimulus, 6-n-propylthiouracil (PROP) [21,28,29,30,31,32,33]. Specifically, volunteers who are very sensitive to PROP (super-tasters) and have a higher fungiform papillae density on their tongues [26,34,35,36,37,38], seem to have a higher sensitivity and a lower preference for high-fat foods [28,30,31,39,40,41,42,43], compared to those who taste PROP only at high concentrations or not at all (non-tasters) and show a higher preference for fat-rich foods. These considerations support the hypothesis that PROP sensitivity is negatively related to calorie consumption and body weight, as several studies have reported [30,41,44,45,46,47,48]. Besides, PROP taster status can affect lipid metabolism in normal weight [49] and obese [50] volunteers. However, the role of the PROP phenotype in fat perception is debatable [51,52,53,54,55,56]. Divergent results may be due to the fact that psychophysical methods which are based on self-reports can produce subjective evaluations.

Based on these statements and given the nutritional value of dietary fats, it is of great importance to characterize factors that may contribute to individual differences in fat taste perception with the aim of better understanding the mechanisms involved in the choice of fat-rich foods. We analyzed the relationships between oleic acid taste perception and *rs1761667* SNP in the *CD36* gene and PROP phenotype by direct evaluation of the degree of activation of the peripheral taste system in response to the oleic acid taste stimulation. These measures were carried out by means of electrophysiological recordings from the human tongue (Electrotastegrams, ETG), which yield data that are not influenced by the individual’s subjective biases [57,58].

## 2. Materials and Methods

### 2.1. Participants

Thirty-five Caucasian and non-smoker volunteers (15 males, 20 females, age 28.6 ± 0.86 years) were recruited according to standard procedures at Cagliari University. All volunteers were originally from Sardinia island, Italy. No statistical analyses were performed to pre-determine the size of the sample. However, several guiding criteria were used. First, our sample size is comparable to those typically employed in electrophysiological recording experiments since they provide a direct measure of the degree of activation of the receptors or neurons under study [57]. Due to the high frequency of AG heterozygotes at the *rs1761667* SNP in the *CD36* gene among American Caucasian [22] and European populations reported in 1000 Genomes (dbSNP Short Genetic Variations, 2017), it was not possible to construct equal sample sizes within each of the genotype/phenotype subgroups. Therefore, volunteers were recruited to form three roughly equal-sized PROP-taster groups that were matched for age and gender. Volunteers had a normal body mass index (BMI) of 20.2 to 25.2 kg/m^2^. None were dieting, taking medications that might interfere with oral sensory perception or had food allergies. Their gustatory function was screened for four basic tastes by a taste strip test (Burghart Messtechnik, Wedel, Germany) to rule out any gustatory impairment. All volunteers were informed about the aim and protocol of the study and signed an informed consent form. The present study was conducted in accordance with the latest revision of the Declaration of Helsinki, and the procedures have been approved by the Ethical Committee of the University Hospital Company (AOU) of Cagliari, Italy. This trial was registered at ClinicalTrials.gov (identifier number: UNICADBSITB-1).

### 2.2. Experimental Protocol

Volunteers were tested in two sessions on two successive days. On the first day, each subject was classified for his/her PROP taster status, while on the second day he/she was tested for the electrophysiological response to oleic acid taste stimulation on a small area of the tongue tip and for the fungiform papillae density on the same area of the tongue surface. Volunteers were always requested to abstain from drinking (except water), eating, using chewing gum or oral care products for at least 2 h prior to testing. All had to be in the test room 15 min before the beginning of the session (9:00 AM) in order to acclimate to the constant environmental conditions (23–24 °C; 40–50% relative humidity). Women were always tested around the sixth day of the menstrual cycle to avoid taste sensitivity changes due to the estrogen phase [59]. All solutions (in spring water), which were used for the measures at room temperature, were prepared and stored in a refrigerator until 1 h before testing. Stimuli were presented.

At the end of the first visit, samples of whole saliva (2 mL) were collected from each volunteer into an acid-washed polypropylene test tube by means of a soft plastic aspirator. The samples were stored at −80 °C until being processed by the molecular analyses described below.

The study design is shown in Figure 1.

### 2.3. PROP Taster Status Classification

Volunteers were classified for their PROP taster status by two scaling measurements. All were first assessed using the three-solution test [60], which has been validated in numerous studies [61,62,63,64,65]. The perceived taste intensity ratings to PROP (0.032, 0.32, and 3.2 mmol/L) (Sigma-Aldrich, Milan, Italy) and sodium chloride (NaCl; 0.01, 0.1, 1.0 mol/L) (Sigma-Aldrich, Milan, Italy) solutions were collected by using the Labeled Magnitude Scale (LMS) [66]. The use of this scale gave the volunteers the freedom to evaluate the PROP bitterness intensity relative to the “strongest imaginable” oral stimulus ever perceived in their life. LMS is a semi-logarithmic 100-mm scale in which seven label verbal descriptors are arranged, in semilog intervals, along the length of the scale. The verbal labels and their positions on the LMS are: barely detectable, 1.4; weak, 6.1; moderate, 17.2; strong, 35.4; very strong, 53.3; and strongest imaginable, 100.

NaCl was used as a control because taste intensity to NaCl does not change with PROP taster status in this procedure [60]. Concentrations (10 mL samples) were presented in a random order. Volunteers who gave lower intensity ratings to PROP than to NaCl were classified as PROP non-tasters, those who gave overlapping ratings to both PROP and NaCl were classified as medium tasters, and those who gave higher ratings to PROP than to NaCl were classified as super-tasters. The classification of each subject as belonging to a PROP taster group (super-taster, medium-taster, or non-taster) was confirmed using the impregnated paper screening test [27,67] after a 1-h period. With this method, the two stimuli were presented sequentially to each subject by placing the paper disks on the tip of the tongue for 30 s; the first one was impregnated with PROP solution (50 mmol/L) and the second with NaCl (1.0 mol/L). The ratings of the perceived intensity on each paper disk were obtained by using the LMS scale (described above). Volunteers who rated the PROP disk lower than 15 mm on the LMS were categorized as non-tasters; those who rated the PROP disk higher than 67 on the LMS were categorized as super-tasters; all others were classified as medium tasters [67]. Volunteers who were classified differently by two methods were excluded from other tests. Based on the classification, which was documented by three-way ANOVA, 12 volunteers (5 males, 7 females, age 26.5 ± 2.6 years) were classified as non-taster (34.29 %), 13 (6 males, 7 females, age 29.2 ± 1.8 years) were medium taster (37.14 %) and 10 (4 males, 6 females, age 25.2 ± 2.6 years) were super-taster (28.57 %) (Appendix A).

### 2.4. Molecular Analysis

Volunteers were genotyped for the *rs1761667* (G/A) single nucleotide polymorphism (SNP) of *CD36*, located at the —31118 promoter region of exon 1A. Briefly, analyses were performed by PCR followed by analysis with restriction enzyme (*Hha*I) of the fragments obtained according to Banerjee et al. 2010 [68]. This method has been validated by numerous studies [21,24,33]. The products of digestion were separated by electrophoresis on a 2% agarose gel and the bands of DNA were visualized by ethidium bromide staining and ultraviolet light to score the deletion. PCR 50 bp Low Ladder DNA was used as a molecular mass marker (Gene Ruler™-Thermo Scientific, Waltham, MA, USA).

Volunteers were also genotyped for the three SNPs of *TAS2R38* (gene which expresses the specific receptor of PROP [69]), that give rise to two major haplotypes, the taster variant (PAV) and the non-taster variant (AVI), and three rare ones (PVI, AAI, and AAV). Molecular analyses of *TAS2R38* locus were performed by Taqman^®^ SNP Genotyping Assays (Applied Biosystems by Life-Technologies Italia, Europe BV) [57].

### 2.5. Electrophysiological Recordings

Differential electrophysiological recordings from the tongues of volunteers were performed according to Sollai et al. 2017 [57]. Briefly, two silver electrodes were used, one in contact with the tongue ventral surface and the other in perfect adhesion with the dorsal surface. The first electrode was a silver wire (0.50 mm) with the distal terminal rolled up to form a ball (about 5 mm of diameter) to obtain a good electrical contact and make the electrode safe for the sublingual mucosa. The second one (patent WO 2017/212377) was made by depositing a silver film (100 nm thick) on a very thin (13 μm) polyimide layer (Kapton ©, DuPont, Wilmington, DE, USA) by means of evaporation in high vacuum. A film of insulating and biocompatible material (Parylene C, 2 µM thick) covered both sides of the electrode except for the area which must be in electrical contact with the tongue. The extreme suppleness of this electrode allows its perfect adhesion with the dorsal surface of the tongue. The distal part of this electrode had a circular hole which leaves a small area (6 mm of diameter) uncovered on the left side of the tongue surface tip when it is positioned on the tongue surface. This is the area of the tongue where oleic acid stimulation was delivered during the electrophysiological recordings, and the fungiform papillae density is calculated as described below. A third disposable adhesive electrode used as the ground terminal of the measuring instrument, was placed in an electrically neutral position (CDES003545, SpesMedica, Italy). After positioning the electrodes and verifying the electrical contact, the bio-potential recording started when a stable baseline was observed. Signals detected by the electrodes were recorded by a high input impedance polygraph for human use (Porti7 portable physiological measurement system; TMS International B.V., The Netherlands), which is an isolated certified Class IIa medical device. Signals were digitized, recorded and visualized in real time on a PC by PolyBench software (TMSI, Oldenzaal, The Netherlands). For each subject, the recording lasted 55 s (20 s baseline, 15 s during oleic acid simulation and 20 s after stimulation). Afterwards, the waveform of bio-potentials was analyzed (Clampfit 10.0 software, Berkeley, CA, USA) and the measures of voltage changes with respect to baseline in response to oleic acid (amplitude values) were determined at 2.5, 5, 10, and 15 s from stimulation onset. The rate of potential variation (mV/s) was also calculated at the same time intervals.

### 2.6. Oleic acid Taste Stimulations

The oleic acid taste stimulation was delivered by placing for 15 s a paper disk (6 mm dia) impregnated with 30 µL of oleic on the circular area of the tongue surface that was left free by the hole of the second electrode. Each volunteer was instructed to rate the perceived intensity by using the LMS scale [66]. Dry paper disks were also used as control.

### 2.7. Density Measurements of Fungiform Papillae

Fungiform papillae density was measured in the small circular area of the left side, close to the midline of the anterior surface of the tip of the tongue where oleic acid stimulation was delivered during the electrophysiological recordings according to our previous work [57]. Briefly, volunteers sat on a chair supporting their head with their hands in order to minimize movements. The area of the tongue was first dried and then stained by placing a circle of filter paper (6 mm in diameter) impregnated with a blue food dye (E133, Modecor Italiana, Italy) on the specified area. Photographs of the stained area of the tongue surface were taken for each volunteer using a Canon EOS D400 (10 megapixels) camera with lens EFS 55–250 mm. The digital images were analyzed using a “zoom” option in the Adobe Photoshop 7.0 program. The fungiform papillae were separately identified and counted by three trained operators who were uninformed of the PROP taster status and *CD36* genotype of volunteers [26,37,57]. The density/cm^2^ was calculated.

### 2.8. Statistical Analysis

Simple linear correlation analysis was used to investigate the relationship between the density of fungiform papillae and perceived intensity in response to oleic acid taste stimulation. Linear correlation analysis was also used to elucidate the relationships between signal amplitude (mV) and biopotential variation rate (mV/s) with density of fungiform papillae, or perceived taste intensity of oleic acid. Fisher’s method (Genopop software version 4.0) [70] was used to test *CD36* genotype distribution and allele frequencies according to PROP taster status. One-way ANOVA was used across PROP taster groups, *TAS2R38* genotype groups, *CD36* genotype groups or gender, to compare mean values ± SEM of the perceived taste intensity, density of fungiform papillae, signal amplitude (mV), and biopotential variation rate (mV/s). Repeated measures ANOVA was used across PROP taster groups or *CD36* genotype groups, to evaluate the differences of mean values ± SEM of the signal amplitude (mV) and biopotential variation rate (mV/s), at 2.5, 5, 10, and 15 s after the application of oleic acid stimulation. Data were verified for the assumptions of homogeneity of variance, normality and sphericity (when applicable). To determine if the sphericity assumption was violated, a Greenhouse–Geisser correction or Huynh–Feldt correction was applied. Post-hoc comparisons were conducted with the Fisher LDS test, unless the assumption of homogeneity of variance was violated, in which case the Duncan’s test was used. Statistical analyses were conducted using STATISTICA for WINDOWS (version 7; StatSoft Inc, Tulsa, OK, USA). *p* values ≤ 0.05 were considered significant.

## 3. Results

### 3.1. CD36 Genotyping and Phenotyping

Linear correlation analysis showed that the density of fungiform papillae in the small circular area of the tongue where oleic acid stimulation was delivered during the electrophysiological recordings is linearly correlated with the perceived taste intensity by volunteers (*r* = 0.477; *p* = 0.005).

Molecular analysis at the CD36 (SNP: *rs1761667*) gene identified 6 AA homozygous (3 males, 3 females, age 29.2 ± 2.9 years), 20 heterozygous (9 males, 11 females, age 26.1 ± 1.5 years), and 9 GG homozygous volunteers (3 males, 6 females, age 29.7 ± 3.4 years). PROP taster groups did not differ statistically based on genotype distribution and haplotype frequency of the *CD36* gene (*χ*^2^ > 0.665; *p* < 0.71; Fisher’s test).

Mean values ± SEM of perceived intensity after taste stimulation with oleic acid in volunteers classified by their PROP taster status and genotyped for the *rs1761667*SNP of *CD36* gene are shown in Figure 2. The perceived intensity ratings to oleic acid stimulation were associated with PROP taster status. Specifically, the perceived intensity was higher in the super-taster volunteers than in non-taster or medium-taster ones (*p* ≤ 0.046; Duncan’s test subsequent to one-way ANOVA; *F*_2,32_ = 3.138; *p* = 0.054). No differences in intensity ratings between the medium-taster and non-taster volunteers were found (*p* > 0.05). In addition, volunteers with the GG genotype gave intensity ratings higher than volunteers with the AA genotype (*p* = 0.047 Fisher LDS test). No differences between the heterozygous and homozygous volunteers were found (*p* > 0.05).

Fungiform papillae density varied with PROP taster status (*F*_2,32_ = 18.712; *p* < 0.001) (Figure 3). Super-tasters showed a higher density than medium-tasters (*p* < 0.001; Duncan’s test), whose values were higher than those of non-tasters (*p =* 0.032; Duncan’s test). No differences in fungiform papillae density related to the *CD36* polymorphism were found (*p* < 0.05).

### 3.2. Electrophysiolgical Responses to Taste Stimulation with Oleic Acid

The electrophysiological recording from the human tongue allowed us to determine bioelectrical potential changes in response to oleic acid taste stimulation. The analysis of the waveform of bioelectrical potentials showed that the oleic acid stimulation evoked positive monophasic potentials characterized by a faster initial rise followed by a slower phase, which continued for the whole duration of stimulation. The variation in voltage with respect to baseline (i.e., the amplitude of these signals, as well as the hyperpolarization rate) was highly variable among volunteers. The values of amplitude varied from a minimum of 0.64 mM (measured in a non-taster volunteer with the AA genotype in the *CD36* polymorphism) to a maximum of 91.99 mV (determined in a super-taster volunteer with the GG genotype in the *CD36* polymorphism). Examples of this variability are shown in Figure 4.

Linear correlation analysis showed that signal amplitude, as well as hyperpolarization rate, were linearly correlated to the density of fungiform papillae (*r* = 0.394; *p* = 0.028 and *r* = 0.410; *p* = 0.019, respectively). No such correlation was found between the two electrophysiological parameters and perceived taste intensity (*r* < 0.251; *p* > 0.06).

Mean values ± SEM of the amplitude and hyperpolarization rate of signals recorded in response to oleic acid taste stimulation in volunteers categorized for their PROP taster status and genotyped for the *rs1761667* SNP of the *CD36* gene are shown in Figure 5. Values determined in super-taster volunteers were significantly higher than those measured in non-tasters, although they are at the limits of statistical significance (*p* = 0.052; Duncan’s test subsequent one-way ANOVA), while medium-tasters showed biopotential changes which were not different from those of the other two taster groups (*p* > 0.05). In addition, the values determined in homozygous GG volunteers were higher than those of volunteers with the AA genotype (*p* = 0.043; Fisher LDS test subsequent one-way ANOVA). No differences between heterozygous and homozygous volunteers were found (*p* > 0.05). Volunteers with the GG genotype also showed higher values of hyperpolarization rate than volunteers with the AA genotype (*p* = 0.028; Fisher LDS test subsequent one-way ANOVA), while heterozygous volunteers did not show different hyperpolarization rate values from the other groups (*p* > 0.05). No differences in the rate values related to PROP taster status were found (*p* > 0.05). One-way ANOVA also showed that no differences in the amplitude and hyperpolarization rate of signals were found in relation to the *TAS2R38* genotype (Appendix A) or gender (*p* > 0.05).

The same data from Figure 2, Figure 3 and Figure 5 are also reported as distributions of the original points and mean values ± SEM for each PROP taster and *CD36* genotype group as shown in Appendix A.

Mean values ± SEM of signal amplitude and hyperpolarization rate, determined after 2.5, 5, 10, and 15 s after the application of oleic acid taste stimulation according to PROP taster status and CD36 polymorphisms, are shown in Figure 6. The time course of the hyperpolarization amplitude and variation rate were different in volunteers with different PROP phenotypes or *CD36* genotypes. In particular, the amplitude of signals increased during the stimulation up to 15 s in super-taster and medium-taster volunteers (*p* < 0.001 and *p* = 0.004; Fisher LDS or Duncan’s test subsequent repeated measured ANOVA), but they did not change in non-tasters (*p* > 0.05). The hyperpolarization values increased to the end of stimulation in volunteers with the GG genotype in the CD36 gene *(p* ≤ 0.001; Fisher LDS or Duncan’s test subsequent repeated measured ANOVA), but only for a duration of 10 s in heterozygous volunteers (*p* = 0.025; Fisher LDS test subsequent repeated measured ANOVA). These values did not change with time in volunteers with the AA genotype (*p* > 0.05). The hyperpolarization rate decreased (at 10 s and 15 s) in super-tasters (*p* ≤ 0.037; Fisher LDS or Duncan’s test subsequent repeated measured ANOVA) and more rapidly in medium-tasters (*p* ≤ 0.019; Fisher LDS or Duncan’s test subsequent repeated measured ANOVA), while it decreased only at 5 s in non-tasters (*p* = 0.049; Fisher LDS test subsequent repeated measured ANOVA). The hyperpolarization rate rapidly decreased across the whole-time course of recordings in volunteers having the GG genotype in the *CD36* gene *(p* ≤ 0.037; Fisher LDS or Duncan’s test subsequent repeated measured ANOVA). The hyperpolarization rate only decreased at 10 s in heterozygous volunteers (*p* = 0.005; Fisher LDS test subsequent repeated measured ANOVA) and did so more slowly (at 15 s) in volunteers with the AA genotype (*p* > 0.05).

Dry paper disks which were used as controls evoked no potential variations.

## 4. Discussion

A great deal of conflicting data has been collected over the last decade on individual differences in fat perception, preference and consumption related to genetic variation in PROP taste sensitivity [23,51,52,53,54,55,56,71]. Emerging evidence suggests that variation in other genes such as polymorphisms in the gene of the CD36 scavenger protein, may also be involved in fat perception [5,20,21]. The present study provides the first direct demonstration of the roles of PROP phenotype and a *CD36* polymorphism in individual variability in fat perception. In fact, the highly reliable ETG method used here, allowed us to obtain direct and quantitative data that are free from individual subjective responses.

We found that oleic acid taste stimulation evoked positive monophasic potentials, which lasted for the entire duration of the stimulation and, in most recordings, even longer. On the other hand, the control stimulations were ineffective in evoking this response. The extended activation that we found in response to oleic acid could reflect the persistence of stimulation over time (a limitation of our method is that the stimulus cannot to be removed), and its slow increase of amplitude could depend on the high surface tension of lipid molecules. The biopotential variations recorded in response to oleic acid possibly represent a measure of the summated voltage change resulting from the response of stimulated taste cells, as already shown in response to other stimuli in our previous study [57]. These variations are also similar to those recorded from the olfactory epithelium [72,73], where the electrical activity has been reported as the summated generated potential by the population of stimulated olfactory neurons [74]. The fact that the recorded signals effectively correspond to the summated response of stimulated taste cells seems to be confirmed by the changes of the amplitude of signals among the PROP taster groups, which vary in density of papillae. This interpretation is further supported by the direct and linear correlation found between the amplitude and rate of signals and density of fungiform papillae. Density of papillae at the tongue tip, which is highly correlated with their total number on the tongue [37], was also positively correlated with the perceived taste intensity by the volunteers.

In addition, our results show a direct relationship between the amplitude of biopotentials recorded and PROP phenotype. Specifically, we recorded the largest amplitude values in PROP super-tasters who had the highest density of fungiform papillae in the same area of the tongue where oleic acid stimulation was delivered during the recordings. Likewise, the smallest amplitude values were recorded in non-tasters who had the lowest density of papillae, and intermediate amplitudes were recorded in medium-tasters with an intermediate density of papillae. These results strongly support previous psychophysical experiments showing a direct relationship between fat perception and PROP taster status, [21,29,30,31,48,75] that can be linked to differences in the density of papillae across the three PROP taster categories [26,34,35,36,37,38]. In fact, no differences in electrophysiological responses to oleic acid were found in relation to *TAS2R38* polymorphisms. This suggests that the phenotypic expression of the trait, which is strongly associated with papillae density, is a critical determinant of electrophysiological responses to fatty acids on the tongue rather than strictly the presence or absence of specific genetic variants in *TAS2R38*. This is consistent with data from another study, showing that the PROP phenotype is a better predictor of adiposity in women than the *TAS2R38* genotype, which is unrelated to adiposity [46]. Importantly, the signal amplitudes that we recorded in the PROP taster groups, as well as in the *CD36* genotype groups, agree with the perceived intensities reported by the volunteers during oleic acid stimulation, thus indicating that our bioelectrical measurements are fully consistent with common human psychophysical observations. The lack of a direct linear correlation between the electrophysiological and psychophysical measurements may simply reflect the presence of background ‘noise’ in the electrophysiological measures, and further refinement of the recording procedure will presumably reduce this noise.

Another important physiological feature we observed was a relationship between the values of the hyperpolarization amplitude and rate, and the *rs1761667* polymorphism of the *CD36* gene. In agreement with evidence showing that the presence of the homozygous AA genotype at this location of the *CD36* gene is characterized by reduced protein expression [19] and low taste sensitivity to fats [5,20,21,22], we measured the lowest amplitude values and biopotential variation rates in volunteers carrying two A alleles who verbally reported the lowest values of perceived intensity. Likewise, we measured the highest bioelectrical values in volunteers with the GG genotype who perceived the highest intensity of oleic acid and intermediate values in heterozygous volunteers who reported intermediate values of perceived intensity. As expected, no variations in the density of papillae related to *CD36* genotypes were found.

The analysis of the time course of the responses showed that the amplitude of the signal increased during stimulation, mostly in the last portion of the response, and this effect was most prominent in PROP tasters who had a higher number of papillae, and in volunteers having at least one G allele in *CD36,* which is known to be associated with an increase of receptor expression [19]. On the contrary, signal amplitude did not change in PROP non-tasters who had a low number of papillae and in volunteers homozygous for the non-tasting (AA) form of this polymorphism in *CD36,* which is associated with reduced protein expression [19]. In addition, the hyperpolarization rate rapidly decreased across the whole time course of recordings in volunteers with two tasting (GG) alleles, who showed at 2.5 s after stimulus onset, values about twice as high as those of the volunteers with only a single G allele. In turn, the hyperpolarization rate slowly decreased in both heterozygous (GA) volunteers and in volunteers with two non-taster variants in *CD36*. All these results suggest that the presence of the tasting variant in the specific receptor is the most important condition to elicit a prompt response and, in addition, turns out to be the most important condition to evoke an intense perception when the volunteers have a high number of fungiform papillae in their tongue.

Future studies should confirm the results in a larger population (with a higher number of subjects in each study groups). In fact, a limitation of this work is the small size of the examined sample mostly regarding the group of subjects with the homozygous AA genotype at this *CD36* locus.

## 5. Conclusions

In conclusion, the present work builds on our previous psychophysical studies documenting a role for *rs1761667* polymorphism in *CD36* and polymorphisms in the *TAS2R38* gene (indexed by PROP phenotype in the current study) in the perception of oleic acid [21]. Here, we used a novel physiological recording technique [57] to directly measure the degree of activation of the peripheral gustatory system in response to taste stimulation with oleic acid on a localized area of the tongue. We found that both genes contributed to variations in the perception of oleic acid, but they had somewhat different effects on specific features of the electrophysiological response. Rate variation seemed to be influenced mostly by the *CD36* gene, early in the time course, whereas signal amplitude was more influenced by PROP status during the latter part of the time course. The influence of PROP status on signal amplitude may reflect a summation effect associated with higher density of papillae, a well-known anatomical characteristic of PROP-tasting volunteers [26,34,35,36,37,38], although other authors did not find links between the density of papillae and PROP status [76]. Our findings support the notion that several overlapping mechanisms are involved in fat perception, one related to PROP status and papillae density, and another related to *CD36* genotypes. Undoubtedly, other gene effects play a role as well. Further investigation of these mechanisms using both electrophysiological and psychophysical methods could shed important light on how dietary fats are perceived and their contribution to food choice and nutritional status. 

## Figures and Tables

**Figure 1 nutrients-11-00315-f001:**
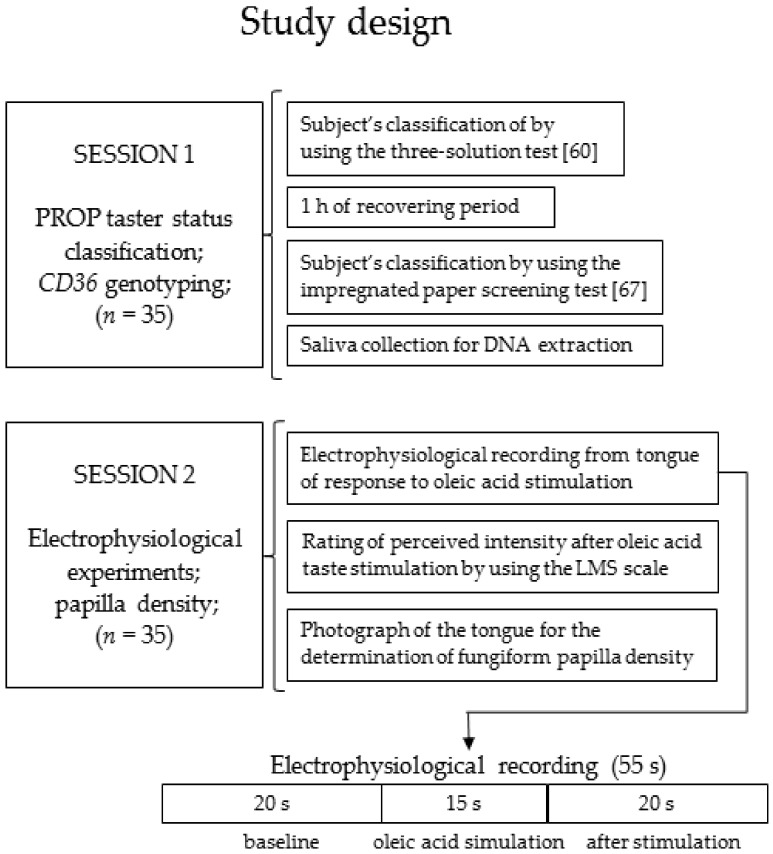
A graphic diagram representing the study design.

**Figure 2 nutrients-11-00315-f002:**
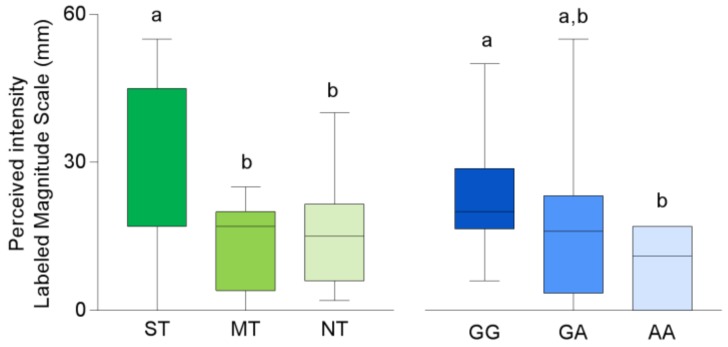
Box-and-whisker plots showing the minimum, first quartile, median, third quartile, and maximum of each set of perceived intensity data evoked by taste stimulation with oleic acid (30 µL) in super-tasters (ST; *n* = 10), medium-tasters (MT; *n*=13) and non-tasters (NT; *n* = 12) and in volunteers with genotypes GG (*n* = 9), GA (*n* = 20) and AA (*n* = 6) of *CD36*. Different letters indicate a significant difference (*p* ≤ 0.046; Duncan’s test and *p* = 0.047 Fisher LDS test, subsequent one-way ANOVA).

**Figure 3 nutrients-11-00315-f003:**
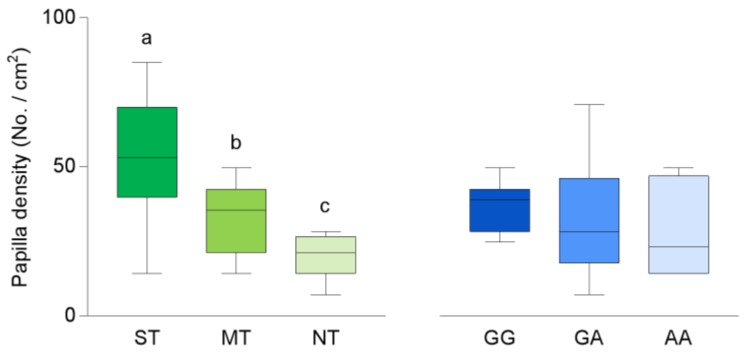
Box-and-whisker plots showing the minimum, first quartile, median, third quartile, and maximum of each set of density of fungiform papillae data in super-tasters (ST; *n* = 10), medium-tasters (MT; *n* = 13) and non-tasters (NT; *n*=12) and in volunteers with genotypes GG (*n* = 9), GA (*n* = 20) and AA (*n* = 6) of *CD36*. Different letters indicate a significant difference (*p* ≤ 0.032; Duncan’s test subsequent one-way ANOVA).

**Figure 4 nutrients-11-00315-f004:**
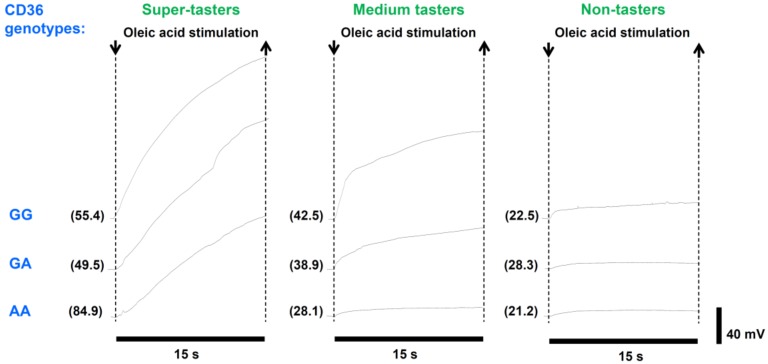
Examples of electrophysiological recordings in response to oleic acid (30 µL) taste stimulation in representative super-tasters, medium-tasters and non-tasters with different genotypes of the *CD36* gene. The very first data point on the left side of each electrophysiological recording represents the baseline. Numbers within parentheses on the left of each trace indicate the density of fungiform papillae (No./cm^2^) of each subject calculated in the small circular area of the tongue where oleic acid stimulation was applied.

**Figure 5 nutrients-11-00315-f005:**
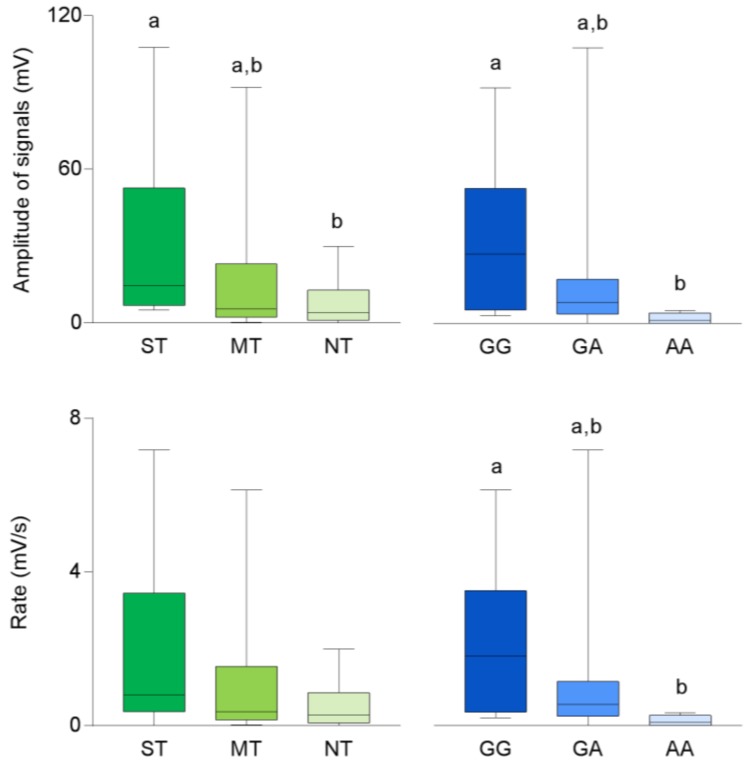
Box-and-whisker plots showing the minimum, first quartile, median, third quartile, and maximum of each data set of amplitude and rate of signals evoked in super-tasters (ST; *n* = 10), medium tasters (MT; *n* = 13) and non-tasters (NT; *n* = 12); and in volunteers with genotypes GG (*n* = 9), GA (*n* = 20) and AA (*n* = 6) of *CD36* by oleic acid (30 µL) taste stimulation. Different letters indicate a significant difference (*p* ≤ 0.05; Fisher LDS or Duncan’s test subsequent one-way ANOVA).

**Figure 6 nutrients-11-00315-f006:**
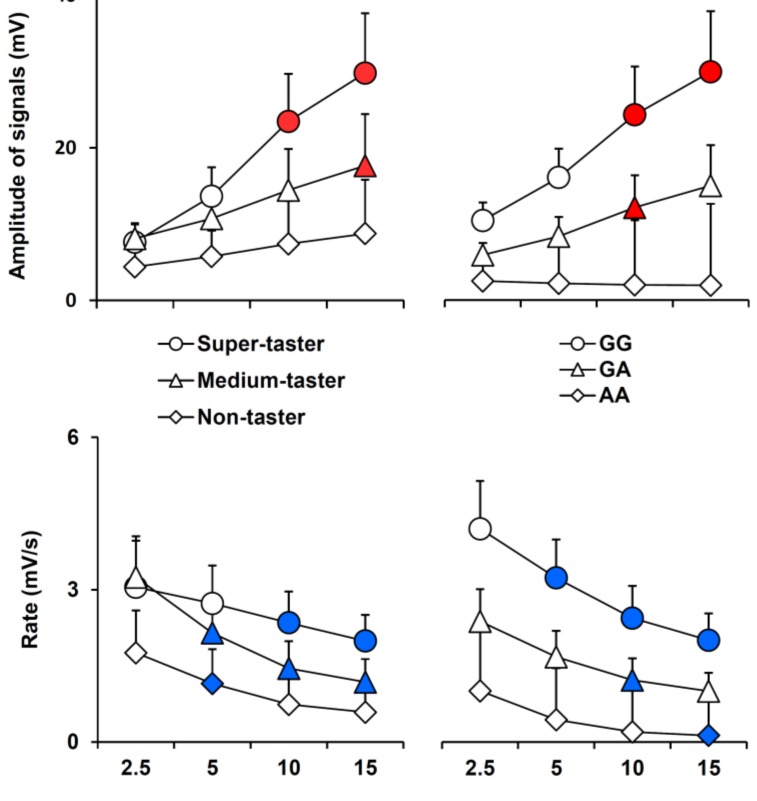
Time course of amplitude (mV) or hyperpolarization rate (mV/s) of the signal across PROP taster status or *CD36* polymorphism groups during stimulation time. Data (mean values ± SEM) are determined after 2.5, 5, 10, and 15 s from the application of oleic acid (30 µL). *n* = 10 super-tasters, *n* = 13 medium tasters and *n* = 12 non-tasters; *n* = 9 volunteers with genotypes GG in CD36, *n* = 20 GA genotypes and *n* = 6 AA genotypes. Solid symbols (red for amplitude of signals and blue for rate) indicate a significant difference with respect to the previous value of the corresponding group (*p* ≤ 0.05; Fisher LDS or Duncan’s test, subsequent to repeated measures ANOVA across PROP taster groups or *CD36* genotype of volunteers).

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
