# Peer review of "Human Tongue Electrophysiological Response to Oleic Acid and Its Associations with PROP Taster Status and the CD36 Polymorphism (rs1761667)"

_nutrients, 2019, doi:10.3390/nu11020315_

Reviewer 1 Report

The manuscript had demonstrated the use of objective response such as electrophysiological response to measure CD36 polymorphism in relation to PROP taster status. The results of this manuscript would further emphasise the understanding of CD36 polymorphism in a more objective manner. 

There are some sections that needs revision on the manuscript that the authors should consider.

Methodology - The authors should justify why the sample number had been chosen. Additionally, it is unclear that the electrophysiology measurements did not measure baseline where usually it is been done and a measure of %change to the baseline is measured. I'd also recommend the authors add a simple flow chart to summarise the protocols/methods that has been done in the manuscript.

Results - consider revising the p value here, I believe .01 is more than sufficient to show the significance. I'd also recommend the authors to run additional analysis (e.g. a simple correlation analysis) to further elucidate the relationship between the results.

Discussion - The discussion provided a good overview of the research topic but had failed to discuss the specifics of the results. For example, there some results that were not significant observed on the in some cases for Figure 5. I think the authors should attempt to discuss why this was the case in the discussion. 

Overall the manuscript had provided empirical evidence and will shows a great contribution in understanding individual differences especially. However I recommend the authors to provide additional information on the protocols and justify the above comments. 

Author Response

Rebuttal to comments of Reviewer 1

We have reworked the manuscript according to the Reviewers’ comments and suggestions.

In the revised manuscript the changes made according to Reviewer 1 are highlighted in red, those according to Reviewer 2 in blue.

 Reviewer #1

The manuscript had demonstrated the use of objective response such as electrophysiological response to measure CD36 polymorphism in relation to PROP taster status. The results of this manuscript would further emphasise the understanding of CD36 polymorphism in a more objective manner.

There are some sections that needs revision on the manuscript that the authors should consider.

Methodology - The authors should justify why the sample number had been chosen. Additionally, it is unclear that the electrophysiology measurements did not measure baseline where usually it is been done and a measure of % change to the baseline is measured. I'd also recommend the authors add a simple flow chart to summarise the protocols/methods that has been done in the manuscript.

 Reply:

We acknowledge that thirty-five subjects is not a large number, however we would like to emphasize to the Reviewer that electrophysiological experiments typically do not test a lot of subjects since they provide a direct measure of the degree of activation of the receptors or neurons under study.  The goal of our technique is to put together scientific rigor with methodological simplicity and non-invasiveness thus obtaining objective and quantitative data on taste sensitivity of patients by electrophysiological recordings of the response of bud cells. Therefore, our technique permits to obtain results which are not affected by the individual’s subjective confounding factors which normally require testing a large number of subjects.

The number of subjects tested in this study has been conditioned by some factors including the number of electrodes we had available for the study. In fact, although we have patented our measuring devices (patent WO 2017/212377), at the moment the device is not available on the market. Therefore, we had to prepare the electrodes in our laboratory by a handicraft procedure (described in the Methods section) which is more expensive and takes a lot of time per each electrode. In addition, the administration of our University does not allow payment of recruited volunteers, thus limiting the availability of subjects to be included in the panel.

However, we managed to recruit an almost equal numbers of NTs, MTs & STs.  It would be impossible to have equal numbers of CD36 genotypes.  Heterozygotes are much more common in our population. In order to have larger numbers in the homozygous groups or more equal numbers per group, we would have to recruit a very large sample size.  For the reasons presented above this was not feasible. 

We agree with the Reviewer that we did not present enough information for the electrophysiology measurements and this resulted in confusion. We added more details to better describe them in the Electrophysiological recordings paragraph (lines 168-170 and 173-176). For more clarity, also two sentences have been added in the Results section (line 262), and in the Fig 4 legend. In addition, we would like to emphasize that in Fig 4 the very first data point on the left side of each electrophysiological recording corresponds to the baseline.

In addition, a graphic diagram representing the study design.  has been added in the manuscript.

Results - consider revising the p value here, I believe .01 is more than sufficient to show the significance. I'd also recommend the authors to run additional analysis (e.g. a simple correlation analysis) to further elucidate the relationship between the results.

Reply: We comply with the Reviewer’s request. Besides, according to the request of Reviewer 2 we provided three decimal places for the p values.

We thank the Reviewer for recommending to run a simple linear correlation analysis to further elucidate the relationship between the results. The analysis suggested by the Reviewer allowed us to provide another viewpoint of the data. Text about this has been added in the Statistical analysis paragraph and the Results and Discussion sections where appropriate.

Discussion - The discussion provided a good overview of the research topic but had failed to discuss the specifics of the results. For example, there some results that were not significant observed on the in some cases for Figure 5. I think the authors should attempt to discuss why this was the case in the discussion.

Reply: We comply with the Reviewer’s request. We better discussed results of figure 5 (now figure 6) at lines 385-391.

Overall the manuscript had provided empirical evidence and will shows a great contribution in understanding individual differences especially. However I recommend the authors to provide additional information on the protocols and justify the above comments.

 Reviewer 2 Report

The submitted manuscript concerns a study of sensory response in 35 subjects, genotyped for PROP taster and CD36 status, and additionally tested for electrophysiological response to oleic acid (OA).  OA produced robust potentials, the size of which correlated with perceived taste intensity from the panelists, and also showed some association (although somewhat complex in nature) with both PROP taster status and CD36 typing.  The paper is well organized, and results are relatively clear, however do suffer from small sample sizes throughout meaning statistical tests are close to the margins.  Several issues would benefit from some attention to improve the report, given below.

Sample size is very small when divided into sub-samples.  As few as 6 participants in a group.  With this in mind, “dynamite plot” style figures are not appropriate, individual points plus mean should be used to display data.  Especially important as amplitude ratings are stated to vary as much as 100x.

Please provide p values to a standard number of decimal places in all cases (3 would seem sufficient).

It is not clear whether the ultimate conclusion to this paper being presented is that PROP tasting status and OA response are related, or that fungi density and OA response are related (shown to strongly correlate).  If PROP taster status is in fact a surrogate here for TB density, a regression against a constant value of TB density would be a more valid analysis strategy than categorical analysis via taster status.  If it is really PROP tasting status that’s important, some logical model should be put forward to suggest why this might be.

Why were the subjects genotyped for CD36 status, but only phenotyped for PROP response?

Statement that this is the first “objective data” seems to dismiss all sensory data as subjective.

Line 35-36 – Very general, what kind of effects?

Likewise line 53, provide detail on ethnic-specific effects

Some reference must be made to the studies that show no link between papilla density and PROP status e.g. Dinella et al, 2018.

The logic suggesting that “These considerations 64 support the hypothesis that PROP tasting is inversely related to calorie consumption” is not clearly expressed.

Line 119 – define similar ratings

Line 126 - please give this figure in % of scale and not in mm

Line 147 – please give city and country for company throughout.

Please provide a breakdown of panel by PROP taster and CD36 typing for age and gender.

Figure 1 (and throughout) – Scale in mm is not too informative, providing label anchors in figure or legend.  Please specify in legend if bars represent SEM?  Please indicate in legend conc of stimulus in this and future legends.

Electrophysiological results from figure 3 and 4 would benefit from being combined to one large figure, this way the reader can see individual responses and averages for the group on the same page.

Line 229 – States mM, and not mV.

Figure 3 – 2 decimal places for fungi density is not appropriate, original data were not taken with this accuracy.

Lines 353/4, what overlapping mechanisms?  Bitter taste?  More detail is needed to support this statement.

Author Response

Rebuttal to comments of Reviewer 2

We have reworked the manuscript according to the Reviewers’ comments and suggestions.

In the revised manuscript the changes made according to Reviewer 1 are highlighted in red, those according to Reviewer 2 in blue.

Reviewer # 2

The submitted manuscript concerns a study of sensory response in 35 subjects, genotyped for PROP taster and CD36 status, and additionally tested for electrophysiological response to oleic acid (OA).  OA produced robust potentials, the size of which correlated with perceived taste intensity from the panelists, and also showed some association (although somewhat complex in nature) with both PROP taster status and CD36 typing.  The paper is well organized, and results are relatively clear, however do suffer from small sample sizes throughout meaning statistical tests are close to the margins.  Several issues would benefit from some attention to improve the report, given below.

Sample size is very small when divided into sub-samples.  As few as 6 participants in a group.  With this in mind, “dynamite plot” style figures are not appropriate, individual points plus mean should be used to display data.  Especially important as amplitude ratings are stated to vary as much as 100x.

Reply: We comply with the Reviewer requests. In the figures the “dynamite plot” style has been changed to the “box-and-whisker plots” which, by showing the minimum, first quartile, median, third quartile, and maximum of each set of data, are more appropriate in the case of data with high variability. We tried to prepare figures showing individual points plus mean, but the results did not look too clear.

Please provide p values to a standard number of decimal places in all cases (3 would seem sufficient).

Reply: We comply with the Reviewer’s request. We provided three decimal places for the p values.

It is not clear whether the ultimate conclusion to this paper being presented is that PROP tasting status and OA response are related, or that fungi density and OA response are related (shown to strongly correlate).  If PROP taster status is in fact a surrogate here for TB density, a regression against a constant value of TB density would be a more valid analysis strategy than categorical analysis via taster status.  If it is really PROP tasting status that’s important, some logical model should be put forward to suggest why this might be.

Reply: We comply with the Reviewer’s request. Also according to the request of Reviewer 1, we carried out a simple linear correlation analysis to further elucidate the relationship between electrophysiological parameters and density of fungiform papillae or perceived taste intensity. The analyses suggested by the Reviewer allowed us to provide another viewpoint of the data. Text has been integrated including this in the Statistical analysis paragraph and the Results and Discussion sections where appropriate.

Why were the subjects genotyped for CD36 status, but only phenotyped for PROP response?

Reply: The aim of our work was to characterize factors that may contribute to individual differences in fat taste perception. To this aim we analyzed the effect on oleic acid perception of a variant of the gene codifying for the specific receptor (CD36) and, since several authors studied the relationship between fat perception and PROP phenotype reporting inconsistent results (Introduction, lines 58-72), we wanted to investigate this relationship by means of an electrophysiological measure of the degree of activation of the peripheral taste system which permits to obtain the direct response of the stimulated taste cells, thus excluding individual’s subjectivity that may act as confounding factor.

In addition, we wish to emphasize the PROP phenotype goes far beyond the simple genotype for the specific receptor (TAS2R38), albeit the allelic diversity in TAS2R38 gene explain most of phenotypic variance (Kim et Science 2003; Bufe et al. Curr Biol 2005). Certainly, papillae density is an important determinant (Melis et. 2013 Pls one; Yeomans et al 2007 P&B; Bartoshuk et al. P&B 1994; Essick et P&B 2013; Shahbake et al. Brain research 2005; Bejac et. P&B 2008). Therefore, we preferred to analyze data for PROP taster groups and not for TAS2R38 genotype groups.   

However, we added the supplemental figure 1 to show electrophysiological measurements in the TAS2R38 genotype groups and reworked the manuscript accordingly (Molecular analysis -lines 147-151, Statistical analysis paragraph -lines 203-204, Results - lines 290-292 and in the Discussion – lines 362-368.

Statement that this is the first “objective data” seems to dismiss all sensory data as subjective.

Reply: We comply with the Reviewer’s request. The words “objective data” have been changed with “direct evidence” in the abstract and with “direct demonstration“ in the Discussion section.

Line 35-36 – Very general, what kind of effects?

Reply: We comply with the Reviewer’s request.  We specify the effects as regulators of lipid and energy metabolism (lines 35-36).

Likewise line 53, provide detail on ethnic-specific effects

Reply: It has shown that CD36 polymorphisms may influence different features of fat perception and effects of this gene may vary across ethnic groups. For example, East Asians with GG genotype gave higher fattiness and creaminess ratings compared to those who had AA genotype.  The same study did not find any effects of this SNP among Caucasian subjects. We better specify at lines 52-54.

Some reference must be made to the studies that show no link between papilla density and PROP status e.g. Dinella et al, 2018.

Reply: We comply with the Reviewer’s request. The contribute by Dinella et al, 2018 has been added in the Discussion section.

The logic suggesting that “These considerations 64 support the hypothesis that PROP tasting is inversely related to calorie consumption” is not clearly expressed.

Reply: We comply with the Reviewer’s request. We better explained the logical consequentiality of the two sentences at lines 66-67.

Line 119 – define similar ratings

Reply: We comply with the Reviewer’s request.  The word “similar” has been changed with “overlapping”

Line 126 - please give this figure in % of scale and not in mm

Reply: we are not sure we understand what the Reviewer is referring to. If the He/She is referring to line 126 (now 132) and the subsequent ones, we would like to specify that these lines describe the procedure of the second method (Zhao et al. 2003) we used to validate the results of classification of subjects for their PROP taster status obtained with the first method (Tepper et al. 2001) we used. We have followed the method published by Zhao et al. 2003 which shows criteria for grouping the subjects based on the cutoffs on the LMS scale in mm, not percentage.  

Line 147 – please give city and country for company throughout.

Reply: We comply with the Reviewer request. We gave company, city and country for Kapton.

Please provide a breakdown of panel by PROP taster and CD36 typing for age and gender.

Reply: We comply with the Reviewer request. The breakdown of panel by PROP taster and CD36 typing for age and gender has been added at lines 136-138 and 222-224. We also analyzed the electrophysiological measurements (amplitude and rate of signals) in relation to gender and found no significant differences. This has been added in the Statistical analysis paragraph (line 204) and in the Results at lines (290-292).

Figure 1 (and throughout) – Scale in mm is not too informative, providing label anchors in figure or legend.  Please specify in legend if bars represent SEM?  Please indicate in legend conc of stimulus in this and future legends.

Reply: We comply with the Reviewer request. We modified figure legends accordingly.

Electrophysiological results from figure 3 and 4 would benefit from being combined to one large figure, this way the reader can see individual responses and averages for the group on the same page.

Reply: We recognize that the combination of the electrophysiological results of Figures 3 and 4 (now figure 4 and 5) would allow the reader to see individual responses and averages for the group on the same page. We tried to condense it all in one big figure, but it was impossible to have the legend in the same page without reducing too much the panels and lose details. Therefore, we would like to keep the two figures as they are.

Line 229 – States mM, and not mV.

Reply: We comply with the Reviewer’s request.  “mM” has been changed to “mV”.

Figure 3 – 2 decimal places for fungi density is not appropriate, original data were not taken with this accuracy.

Reply: We comply with the Reviewer’s request. One decimal place for papilla density has been showed in figure 3 (now figure 4).

Lines 353/4, what overlapping mechanisms?  Bitter taste?  More detail is needed to support this statement.

Reply: We comply with the Reviewer’s request. A sentence has been added (lines 412-414) to specify what mechanisms are overlapping.

 Round  2

Reviewer 1 Report

I'd like to thank the authors for addressing the comments. 

Just one small minor comment, perhaps the authors should point out the small population as a limitation in the study in the Conclusion section based on their previous explanation. 

Author Response

We have reworked the manuscript according to the Reviewers’ comments and suggestions.

In the revised manuscript the changes made according to Reviewer 1 are highlighted in blue, those according to Reviewer 2 in red.

Reviewer 1

Just one small minor comment, perhaps the authors should point out the small population as a limitation in the study in the Conclusion section based on their previous explanation.

Reply: We comply with the Reviewer’s request. The following explanations have been added in the Methods (lines 86-93): “However, several guiding criteria were used. First, our sample size is comparable to those typically employed in electrophysiological recording experiments since they provide a direct measure of the degree of activation of the receptors or neurons under study [57]. Due to the high frequency of AG heterozygotes at the rs1761667 SNP in the CD36 gene among American Caucasian [22] and European populations reported in 1000 Genomes (dbSNP Short Genetic Variations, 2017), it was not possible to construct equal sample sizes within each of the genotype/phenotype subgroups. Therefore, volunteers were recruited to form three roughly equal-sized PROP-taster groups that were matched for age and gender”.

In addition, the following sentences have been added in the Discussion section (lines 407-409): “Future studies should confirm the results in a larger population (with a higher number of subjects in each study groups). In fact, a limitation of this work is the small size of the examined sample mostly regarding the group of subjects with homozygous AA genotype at this CD36 locus.

Reviewer 2 Report

While we appreciate the changes the authors have made, a number of our suggestions are not followed, leaving some of the same problems as with the first version.  The key issues that I still have with the MS are the following:

· The original review asked for p values to 3 sig figures.  The primary reason for this was that while a lot were listed to many more than 3 sig. figs, 2 of the most important findings of the paper were just given as “p = 0.05”.  This was worrying to me in the original MS, and in the new one all other p values seem to have been updated to 3 sig figs, EXCEPT for these 2 findings, that are both still listed as “p = 0.05”.  Are readers to assume both happened to be 0.050 exactly?  If so, this is a big coincidence, but then why isn’t this stated as 0.050?  Likewise, a few p values are stated as exact values, and others for example as “p ≤ 0.037” and “p ≤ 0.019”, from the same statistical test.

· In the last review I pointed out that dynamite plots are not suitable for sample sizes as small as 6.  Plots are now changed to another inappropriate style for this sample size (to display quartiles for 6 people means the bottom 1.5 people, next 1.5 people, etc).  With this few data points, we need to see the original points.

· Ratings on the LMS are still given in mm, and not in % of scale.  There is a reason I suggested this change.  As the readership of Nutrients are likely not familiar with this particular sensory scaling technique, % of scale would actually let people know something about the scale.  From the info given in this MS, the reader doesn’t know how big the scale actually is.  You give no info on where the scale ends, or the anchors on the scale, or their relative positions, except the top anchor.  This means a reader not working in sensory science is left without any idea what for instance “15mm on the LMS” actually represents.  The scale’s total size needs to be defined, and the verbal anchors and their relative positions need to be explained in the methods.

Author Response

We have reworked the manuscript according to the Reviewers’ comments and suggestions.

In the revised manuscript the changes made according to Reviewer 1 are highlighted in blue, those according to Reviewer 2 in red.

Reviewer # 2

While we appreciate the changes the authors have made, a number of our suggestions are not followed, leaving some of the same problems as with the first version.  The key issues that I still have with the MS are the following:

·The original review asked for p values to 3 sig figures.  The primary reason for this was that while a lot were listed to many more than 3 sig. figs, 2 of the most important findings of the paper were just given as “p = 0.05”.  This was worrying to me in the original MS, and in the new one all other p values seem to have been updated to 3 sig figs, EXCEPT for these 2 findings, that are both still listed as “p = 0.05”.  Are readers to assume both happened to be 0.050 exactly?  If so, this is a big coincidence, but then why isn’t this stated as 0.050?  Likewise, a few p values are stated as exact values, and others for example as “p ≤ 0.037” and “p ≤ 0.019”, from the same statistical test.

Reply: We comply with the Reviewer’s request. This was an oversight. The two “p = 0.05” has been correct.

On the other hand, we have indicate exact values of p (i.e., “p = …”) when referring at only one value, whereas we indicate “p ≤ …” when we want to refer to the highest value of the various significant differences, when several are present. For example: “the hyperpolarization rate decreased (at 10 s and 15 s) in super-tasters (p ≤ 0.037)”.

· In the last review I pointed out that dynamite plots are not suitable for sample sizes as small as 6.  Plots are now changed to another inappropriate style for this sample size (to display quartiles for 6 people means the bottom 1.5 people, next 1.5 people, etc). With this few data points, we need to see the original points.

Reply: We apologize but we believe firmly that the best way to express data which have a large variability (as we stated at line 267) is to represent them with Box-and-whisker plots, therefore we would rather leave them within the manuscript. However, since the Reviewer wants that the original points are showed, we added the figures with them in the supplementary material.

In addition, we added the following sentences in the Discussion section (lines 407-409): “Future studies should confirm the results in a larger population (with a higher number of subjects in each study groups). In fact, a limitation of this work is the small size of the examined sample mostly regarding the group of subjects with homozygous AA genotype at this CD36 locus.”.

· Ratings on the LMS are still given in mm, and not in % of scale. There is a reason I suggested this change.  As the readership of Nutrients are likely not familiar with this particular sensory scaling technique, % of scale would actually let people know something about the scale. From the info given in this MS, the reader doesn’t know how big the scale actually is. You give no info on where the scale ends, or the anchors on the scale, or their relative positions, except the top anchor. This means a reader not working in sensory science is left without any idea what for instance “15mm on the LMS” actually represents. The scale’s total size needs to be defined, and the verbal anchors and their relative positions need to be explained in the methods.

Reply: We comply with the Reviewer’s request. The following sentences have been added in the methods (lines 126-129): LMS is a semi-logarithmic 100-mm scale in which seven label verbal descriptors are arranged, in semilog intervals, along the length of the scale. The verbal labels and their positions on the LMS are: barely detectable, 1.4; weak, 6.1; moderate, 17.2; strong, 35.4; very strong, 53.3; strongest imaginable, 100.”

Round  3

Reviewer 2 Report

The new version now provides the data originally requested, and in fact invalidates 2 of the major conclusions from the paper.  Despite this, the author's asserted conclusions remain unchanged, now with clearly inaccurate statements, such as “Values determined in super-taster volunteers were significantly higher than those measured in non-tasters”; and “the perceived intensity was higher in super-taster volunteers than in non-taster or medium taster ones”, despite this being totally at odds with their data, or the author’s earlier statement “P values ≤ 0.05 were considered significant.” 

Author Response

The new version now provides the data originally requested, and in fact invalidates 2 of the major conclusions from the paper. Despite this, the author's asserted conclusions remain unchanged, now with clearly inaccurate statements, such as “Values determined in super-taster volunteers were significantly higher than those measured in non-tasters”; and “the perceived intensity was higher in super-taster volunteers than in non-taster or medium taster ones”, despite this being totally at odds with their data, or the author’s earlier statement “P values ≤ 0.05 were considered significant.”

Reply: The p values relative to the sentence “the perceived intensity was higher in super-taster volunteers than in non-taster or medium taster onesare p ≤ 0.046 (240). So, they are significant if P values ≤ 0.05 were considered significant. In the current version we changed “p ≤ 0.046” with “p = 0.046 or p = 0.031”, both significant values.

On the other hand, the sentence Values determined in super-taster volunteers were significantly higher than those measured in non-tasterswas changed as follows: “Values of amplitude determined in super-taster volunteers were higher than those measured in non-tasters, although at the limits of statistical significance (p = 0.052; Duncan’s test subsequent one-way ANOVA)” (lines 284-285).

In addition, we re-iterate that this difference is at the limit of statistical significance in the Discussion section at line 356. We believe these wording changes accurately characterize the strength of these statistical differences. There is no attempt on the part of the authors to understate, obscure or otherwise mischaracterize these findings.

We do not agree with the reviewer’s view that the two findings he/she continues to question (related to PROP taster status), are the two major findings of this study. We emphasize throughout the paper that more than one mechanism is involved in oleic acid perception, and the two mechanisms we investigated (rs1761667 polymorphism in CD36 and PROP phenotype) appear to be overlapping. We showed that both factors contributed to variations in the perception of oleic acid, but they had different effects on specific features of the electrophysiological response. Clearly, these two findings are not as statistically robust as some of the other findings in our study. However, we do not believe this outcome invalidates the overall conclusions of this study. As stated above, we have characterized these data carefully and accurately in both the results narrative and the Discussion and we do not believe we have overinterpreted them. We believe these findings should be interpreted in the context of all the data from the study and not be overemphasized.

We have already acknowledged that a limitation of the study is the small sample size, according to Reviewer 1 request. For this reason, we had already added the following sentences in the Discussion section (lines 407-409): “Future studies should confirm the results in a larger population (with a higher number of subjects in each study groups). In fact, a limitation of this work is the small size of the examined sample mostly regarding the group of subjects with homozygous AA genotype at this CD36 locus” and an explanation of this in the Methods (lines 86-93): “However, several guiding criteria were used. First, our sample size is comparable to those typically employed in electrophysiological recording experiments since they provide a direct measure of the degree of activation of the receptors or neurons under study [57]. Due to the high frequency of AG heterozygotes at the rs1761667 SNP in the CD36 gene among American Caucasian [22] and European populations reported in 1000 Genomes (dbSNP Short Genetic Variations, 2017), it was not possible to construct equal sample sizes within each of the genotype/phenotype subgroups. Therefore, volunteers were recruited to form three roughly equal-sized PROP-taster groups that were matched for age and gender”.